# Validation of Enzyme Immunoassays via an Adrenocorticotrophic Stimulation Test for the Non-Invasive Quantification of Stress-Related Hormone Metabolites in Naked Mole-Rats

**DOI:** 10.3390/ani13081424

**Published:** 2023-04-21

**Authors:** Tshepiso Lesedi Majelantle, Nigel Charles Bennett, Stefanie Birgit Ganswindt, Daniel William Hart, Andre Ganswindt

**Affiliations:** Mammal Research Institute, Department of Zoology and Entomology, University of Pretoria, Pretoria 0028, South Africa; ncbennett@zoology.up.ac.za (N.C.B.); stefanie.ganswindt@up.ac.za (S.B.G.); daniel.hart@zoology.up.ac.za (D.W.H.); aganswindt@zoology.up.ac.za (A.G.)

**Keywords:** adrenocorticotrophic challenge, African mole-rats, Bathyergidae, cortisol, dose-dependent response, glucocorticoids, urine, feces

## Abstract

**Simple Summary:**

Naked mole-rats (*Heterocephalus glaber*) have unique physiological and behavioral characteristics that make them a species of interest in biomedical research. However, due to their small size, frequent blood sampling to monitor the animals is not possible. Thus, we aimed to validate enzyme immunoassays, tools which detect the quantity of hormone molecules, for monitoring stress-related hormones, glucocorticoids, and/or their metabolites, in naked mole-rats using urine and feces. We administered a high and low dose of adrenocorticotrophic hormone, which causes the animal to physiologically respond to stress, and a saline administration as a control, which should not elicit a physiological stress response. The results revealed two enzyme immunoassays, namely, a 5α-pregnane-3β,11β,21-triol-20-one detecting glucocorticoid metabolites with a 5α-3β-11β-diol structure and an 11-oxoaetiocholanolone detecting glucocorticoid metabolites with a 5β-3α-ol-11-one structure, for measuring stress-related hormones in male and female urine, respectively. Furthermore, one enzyme immunoassay, namely, 11-oxoaetiocholanolone detecting 11,17 dioxoandrostanes for measuring stress-related hormones in feces of both sexes. There were sex-related differences in how the animals responded to the high and low doses of the adrenocorticotrophic hormone. We recommend using feces to monitor stress-related hormones in naked mole-rats, which will be valuable when investigating housing conditions and other animal welfare aspects.

**Abstract:**

Small size in mammals usually restricts long-term, frequent monitoring of endocrine function using plasma as a matrix. Thus, the non-invasive monitoring of hormone metabolite concentrations in excreta may provide an invaluable approach. The aim of the current study was to examine the suitability of enzyme immunoassays (EIAs) for monitoring responses to stressors in the naked mole-rat (*Heterocephalus glaber*, NMR) using urine and feces as hormone matrices. A saline control administration, and a high- and low-dose adrenocorticotropic hormone (ACTH) challenge were performed on six male and six female disperser morph NMRs. The results revealed that a 5α-pregnane-3β,11β,21-triol-20-one EIA detecting glucocorticoid metabolites (GCMs) with a 5α-3β-11β-diol structure is the most suitable assay for measuring concentrations in male urine samples, whereas an 11-oxoaetiocholanolone EIA detecting GCMs with a 5β-3α-ol-11-one structure appears the most suitable EIA for quantifying GCMs in female urine. An 11-oxoaetiocholanolone EIA detecting 11,17 dioxoandrostanes was the most suitable EIA for quantifying GCMs in the feces of both sexes. There were sex-related differences in response to the high- and low-dose ACTH challenge. We recommend using feces as a more suitable matrix for non-invasive GCM monitoring for NMRs which can be valuable when investigating housing conditions and other welfare aspects.

## 1. Introduction

The naked mole-rat (*Heterocephalus glaber*) is a monotypic genus in the subterranean African mole-rat family, the Bathyergidae, and is widely distributed across the horn of Africa including in Somalia, Kenya, Ethiopia, and Djibouti [1]. It is one of only two mammal species that have been identified as being eusocial [1]. Due to the naked mole-rat’s unique characteristics, such as their social system, extended longevity, cancer resistance, hypoxia, and pain tolerance [2], the species is often housed in captivity for biomedical research globally. As such, the continuous monitoring of their general health and welfare is essential. However, due to their small size, blood sampling restricts long-term and frequent monitoring for in-depth stress-related hormone welfare questions [3].

Quantifying glucocorticoids (commonly cortisol or corticosterone) or their metabolites is useful as a tool to investigate the level of physiological stress and related welfare issues experienced by animals. Herein, stress is defined as a general syndrome occurring in response to any stimulus (stressors) that threatens or appears to threaten the homeostasis of an individual [4,5]. Glucocorticoids (GCs) are part of multiple physiological processes which maintain homeostasis [6]; however, when an animal is confronted with a stressor, the hypothalamic–pituitary–adrenocortical (HPA) axis is activated as a primary response, which results in, among others, an increased secretion of GCs [7]. High levels of GCs stimulate changes in the physiology of an individual, which help it to cope and respond to the stressor [5,7]. Following the GCs’ physiological and behavioral response, circulating glucocorticoids are metabolized by the liver and often excreted as conjugates, via the kidneys into the urine, or via bile into the gut [8,9]. Although beneficial in the short-term to challenge an imminent threat, long-term exposure to elevated levels of GCs can have adverse effects such as immunosuppression, atrophy of tissues, decreased reproductive capacity, and impaired growth [10,11]. In addition, long-term exposure to stressors may lead to HPA exhaustion [12], which in turn can lead to a reduction or no response to varying levels of perceived stressors [13].

There are multiple matrices that can be used for quantifying glucocorticoids or their metabolites such as plasma, feces, urine, hair/feathers, milk, saliva, and eggs [7,14,15]. Feces and urine are often utilized because sampling is feedback-free and seldom requires handling [7,16]. In addition, using feces and urine as a hormone matrix over plasma or serum is especially advantageous in small mammals due to restrictions in frequent blood sampling [3]. Over the last 40 years, studies have examined glucocorticoid metabolite patterns in relation to various potential stressors, e.g., the effect of reproductive activity [17,18], land transformation [19,20], and captivity [21] in various species.

Although monitoring glucocorticoid metabolite concentrations non-invasively in matrixes such as urine and feces is a powerful tool, there can be dietary, species- and sex-specific differences in hormone metabolism and composition [22,23]. In addition, sex and individual differences in response to particular stressors may result from variation in the severity of the stress response, and the duration of the response [24,25]. Thus, the approach for monitoring hormone metabolite concentrations in feces and urine for a species studied for the first time must be reliably validated [26]. One method used for validating enzyme immunoassays is the adrenocorticotropic hormone stimulation (ACTH) test, which is a physiological validation, stimulating glucocorticoid output [7]. In addition, administering different doses of ACTH can be useful when investigating the ability of the enzyme immunoassays to detect a response in relation to the severity of a stressor [3]. The aim of this study was to examine the suitability of five enzyme immunoassays for determining stress-related alterations in glucocorticoid metabolite concentrations in the naked mole-rat using urine and feces as hormone matrices by performing an adrenocorticotropic hormone stimulation test (ACTH challenge). Saline was injected as a control and two doses of synthetic ACTH were administered to examine the influence of dose-dependent responses and handling in male and female dispersal morph naked mole-rats.

## 2. Materials and Methods

### 2.1. Study Species

Twelve captive-bred naked mole-rats (six males and six females) housed at the Department of Zoology and Entomology, University of Pretoria, were involved in this study (Appendix A). These animals were healthy adults which were previously separated as disperser morphs based on body mass (42.4 ± 2.142 g) and frequency of escaping attempts from the tunnel system [27]. These disperser morph naked mole-rats were kept in individual clear plastic chambers (length = 35 cm, width = 30 cm, and height = 20 cm) with nesting material comprised of sterilized wood shavings (Figure 1). The plastic chambers also included a small black plastic nest box with one opening on the side (length = 15 cm, width = 10 cm, and height = 5 cm) that the animal could use as a refugium. The experimental room was maintained on a 12L:12D light schedule and a temperature and humidity range of between 29–30 °C and 40–60%, respectively. The animals were fed chopped vegetables (sweet potatoes, cucumber, apples, and bell peppers) ad libitum every second day. No free water was provided, as the animals derive all water from the food resource [28].

### 2.2. Saline and ACTH Administration

A saline and a synthetic ACTH (Synacthen^®^ depot, Novartis, South Africa (Pty) Ltd.) solution were administered to the animals. There were two administration days; the first administration day (which was five days after the start of sample collection), all animals were intramuscularly injected with 200 µL of sterile physiological saline solution. Subsequently, all animals were randomly selected to receive either a low (20 ug/100 g body mass) or a high (60 ug/100 g body mass) dose of a synthetic ACTH solution 25 days later. The pre-calculated volume of synthetic ACTH was added to sterile physiological saline to result in a total of 200 µL for injection (Appendix A). To establish individual baseline urine glucocorticoid metabolite (uGCM) and fecal glucocorticoid metabolite (fGCM) concentrations, urine and fecal samples were collected for 5 days prior to and for 5 days post-saline and ACTH administration (thus, 10 days’ continuous sample collection), respectively.

### 2.3. Sample Collection

All voided fecal and urine samples were collected between 08:00 and 15:00 daily over the entire experimental period of 35 days (12th November to 16th December 2020). At the beginning of the day, the animal was temporarily removed from the enclosure, and wood shavings were removed to avoid sample contamination and absorption. Thereafter, the enclosure was wiped down with 70% ethanol. The temporary handling was part of each animals’ cleaning routine and was performed as quickly as possible (≤5 min). At the end of each day, wood shavings were returned to the enclosures without removing the animal. On the two administration days (17th November and 11th December 2020), injections started at 10 am, and all produced urine and feces were collected up to 30 h post-injection. The animals were checked every 30 min for freshly voided urine samples which were collected using plastic pipettes labelled with individual animal ID. In addition, all freshly voided fecal samples (entire defecated sample) were collected using forceps. Individually labelled pipettes were rinsed with distilled water, and air-dried between samples whereas forceps were cleaned with ethanol (70%) and air-dried between each sample collection. Samples where feces and urine came into contact were discarded. Overall, 111 urine samples (2–14 samples per animal, total samples from males = 52, total samples from females = 59 samples) and 87 urine samples (1–14 samples per individual, total samples from males = 36, total samples from females = 51) were collected during the ACTH administration and saline administration period, respectively. In addition, 217 fecal samples (6–36 samples per animal, total samples from males = 109, total samples from females = 108) and 204 fecal samples (6–31 per individual, total samples from males = 100, total samples from females = 104) were collected during the ACTH administration and saline administration period, respectively. The collected material was placed in individually labelled plastic vials, and frozen immediately upon collection at −20 °C.

### 2.4. Fecal Steroid Extraction

Fecal samples were freeze-dried and subsequently pulverized using a mortar and pestle [29]. Due to the small sample mass, samples weighing between 0.0150–0.0249 g, 0.0250–0.0369 g, and 0.0370–0.0550 g were extracted using 0.5 mL, 1 mL, and 1.5 mL of 80% ethanol, respectively [30]. Samples less than 0.0150 g were considered too small [31] and not used in any further analyses (*n* = 30). In addition, samples from the same individual collected within 30 min of each other were combined (*n* = 17). Thereafter, suspensions were vortexed for 15 min and subsequently centrifuged at 1500× *g* for 10 min and the supernatant (fecal extracts) formed was transferred into microcentrifuge tubes and stored at −20 °C for further analysis [32].

### 2.5. Enzyme Immunoassays

Defrosted urine samples and fecal steroid extracts collected during the ACTH administration period (6–16 December 2020, 10 days total) were analyzed for immunoreactive uGCM and fGCM concentrations according to the procedure described by Ganswindt [33] using 5 enzyme immunoassays (EIAs). The EIAs tested were: (i) a corticosterone, (ii) a cortisol, (iii) an 11-oxoaetiocholanolone (detecting 11,17-dioxoandrostanes; hereon referred to as 11-oxoaetiocholanolone I), (iv) an 11-oxoaetiocholanolone (detecting fGCMs with a 5β-3α-ol-11-one structure; hereon referred to as 11-oxoaetiocholanolone II), and (v) a 5α-pregnane-3β,11β,21-triol-20-one (detecting fGCMs with a 5α-3β-11β-diol structure). Assay characteristics including antibody cross-reactivities are provided for the cortisol, corticosterone, and 11-oxoaetiocholanolone I EIAs by Palme and Möstl [34], the 11-oxoaetiocholanolone II EIA by Möstl et al. [35], and the 5α-pregnane-3β,11β,21-triol-20-one EIA by Touma et al. [36].

Defrosted urine samples (collected during the saline administration period (12–22 November 2020, 10 days total) were analyzed for immunoreactive uGCM using 5α-pregnane-3β,11β,21-triol-20-one and 11-oxoaetiocholanolone II EIA for males and females, respectively (see Section 3). Similarly, fecal samples collected during the saline administration period were analyzed for immunoreactive fGCM using the oxoaetiocholanolone I EIA for both sexes (see Section 3).

Creatinine concentrations in all native urine samples were determined by a modified Jaffe reaction [37,38]. Samples with creatinine concentrations of less than 0.05 mg/mL were considered too diluted and not used in any further analyses (*n* = 48). All urine steroid concentrations were expressed as mass per mg creatinine (Cr). All fecal steroid concentrations were expressed per mass of dry fecal matter.

The EIA sensitivities and intra-assay and inter-assay coefficients of variation in high- and low-value quality controls for all EIAs are provided in Appendix A. Serial dilutions of urine samples or fecal extracts for the selected EIAs resulted in displacement curves that were parallel to the respective standard curves and had relative variation in the slopes of respective trend lines of <2% for oxoaetiocholanolone I, <2% for oxoaetiocholanolone II EIA, and <2% for 5α-pregnane-3β,11β,21- triol-20-one EIA. All EIA analyses were conducted at the Endocrine Research Laboratory, University of Pretoria, South Africa.

### 2.6. Data Analysis

To identify the suitable EIA for each matrix and sex, firstly, the baseline for each individual and each EIA was calculated. Following from the results of Edwards et al. [16], and due to the limited number of urine samples collected, the median of all samples prior to the injection (hours −120 to 0), and 48 h post-injection until the end of collection (hours 48–120) were used to calculate baselines for urine samples of both sexes. Individuals with sample sizes less than 3 for either ACTH administration and saline administration were considered to have a small sample size and were thus removed from further analysis. Thus, one male (number of samples during ACTH administration period = 8, saline administration period = 1) and one female (number of samples during ACTH administration period = 2, saline administration period = 3). Since there was no decrease in fGCM concentrations 48–120 h post-injection, the baseline fGCM concentrations were calculated from only prior to the injection samples (hours -120 to 0). Thereafter, the post-injection increase was calculated using the difference between the sample concentrations and respective median baseline concentrations as a percentage of the respective median baseline concentration. For each individual, EIA, and matrix, the peak sample within 48 h of administration and with the highest percentage change was identified. Suitability of an EIA was agreed on if the highest percentage change was ≥100% above the individual baseline value for each matrix and sex.

Results from high or low ACTH administration and saline administration (collectively referred to as doses) for each sex were used to investigate the influence of dose-dependent responses. Firstly, individual uGCM and fGCM percentage change data were pooled using the median into baseline, 12-, 24-, 48-, 72-, 96-, and 120-h time intervals for each dose and sex. The time intervals were treated as categorical variables (urine samples included a 6 h time point: Table 1). The percentage increases for both matrixes had a positively skewed distribution, and thus were log_10_ transformed prior to statistical analysis. In addition, the sexes were analyzed separately. A linear mixed-effects model, with ID as a random effect, was used to analyze if there was a significant change in uGCM or fGCM concentrations in relation to the interaction between time interval and dose using the *lme4* package [39]. Normality was tested on model residuals using a Levene’s test, and visually using quantile comparison plots. The results are presented as mean ± standard error (SE) and differences between data sets were found to be significant at *p* < 0.05. All statistical analyses were carried out using R statistical software with RStudio (2022.07.1) interface [40].

## 3. Results

### 3.1. EIA Validation

Of the five EIAs tested for quantifying uGCMs, three EIAs, the cortisol EIA (mean ± SE = 564 ± 304%), corticosterone EIA (234 ± 128%), and 5α-pregnane-3β,11β,21-triol-20-one EIA (198 ± 48%), had an overall percentage increase above 100% for the males within a period of 48 h post-injection (Table 1). On an individual level, two out of five, three out of five, and all the males assessed had an overall percentage increase above 100% post-ACTH injection in the cortisol EIA, corticosterone EIA, and the 5α-pregnane-3β,11β,21-triol-20-one EIA, respectively (Table 2). Consequently, the 5α-pregnane-3β,11β,21-triol-20-one had less variation between the individual’s overall percentage increase (standard deviation = 109%) compared to the cortisol EIA (standard deviation = 681%) and corticosterone EIA (standard deviation = 285%). Thus, the 5α-pregnane-3β,11β,21-triol-20-one EIA was selected as the best-performing EIA for the male urine samples. For the female urine samples, the oxoaetiocholanolone I EIA (71 ± 34%) and oxoaetiocholanolone II EIA (53 ± 33%) showed an overall peak post-ACTH injection within 48 h of ACTH administration (Table 1). However, only two of the five females showed an increase greater than 70% within 48 h post-injection for all EIAs except the 5α-pregnane-3β,11β,21-triol-20-one EIA (Table 1). As the 11-oxoaetiocholanolone II EIA revealed peaks for both females treated with the high dose (Table 1), this EIA was selected for further analyzing the female urine samples. However, based on the current findings, any of the tested EIAs could have been selected, except the 5α-pregnane-3β,11β,21-triol-20-one EIA.

Of the five EIAs tested for fGCM quantification, the oxoaetiocholanolone I was identified as the most suitable EIA for both the male and female naked mole-rats, with the highest peak fGCM concentrations occurring within 48 h post-injection (214 ± 41% and 187 ± 49%, respectively: Table 2). However, one male and two females had a percentage increase less than 100% within 48 h post-injection (Table 2). Even so, the oxoaetiocholanolone I EIA was selected as the best-performing EIA for quantifying the fGCMs in the male and female naked mole-rats.

### 3.2. Dose-Dependent Response in uGCM Concentrations

The linear mixed-effects model for the uGCMs of the males explained 47% (conditional R^2^) of the variation in the percentage change in the uGCM concentrations and the interaction between the time and dose explained 43% (marginal R^2^) of the variation in the percentage change in the uGCM concentrations. The effect of the time period on the percentage change in the uGCM concentrations was dose-dependent (F = 2.590, df = 16, *p* < 0.001: Figure 2). The high-dose treatment (203.68 ± 91.17) had a higher percentage increase in the uGCM concentrations than the low dose (132.33 ± 80.47) at 6 h post-administration (Figure 2A). The saline administration revealed an overall peak (99.09 ± 77.84) at 12 h post-administration (Figure 2A).

The linear mixed-effects model for the female percentage change in the uGCMs explained 30% (conditional R^2^) of the variation in the uGCM concentrations, and the interaction between the time and dose explained 28% (marginal R^2^) of the variation in the percentage change in the uGCM concentrations. The effect of the time period on the percentage change in the uGCM concentrations was not dose-dependent (F = 1.143, df = 20, *p* = 0.296: Figure 2B). However, the high-dose treatment had a peak at 12 h (150.44 ± 79.56) post-injection (Figure 2B). Overall, for the respective EIA, the females had a percentage response lower than 100% for the low dose and saline administration. Interestingly, there was a variation in the raw baseline and response uGCM concentrations between individuals (Figure 3); however, individual variation explained 4% and 2% of the percentage change in the male and female uGCM concentrations, respectively.

### 3.3. Dose-Dependent Response in fGCM Concentrations

The linear mixed-effects model for the male percentage change in the fGCMs explained 33% (conditional R^2^) of the variation in the fGCM concentrations, whereas the dose, time, and their interaction explained 27% (marginal R^2^) of the variation in the fGCM concentrations. The effect of the time periods on the percentage change in the fGCM concentrations was dose-dependent (F = 1.665, df = 17, *p* = 0.042). The low-dose treatment had a peak at 12 h (207.36 ± 107.70) post-injection (Figure 4A). The high-dose treatment led to a peak at 24 h (126.80 ± 53.95) post-injection (Figure 4A). Saline administration did not lead to a peak value at any time point (Figure 4).

For the female percentage change in the fGCMs, the linear mixed-effects model explained 68% (conditional R^2^) of the variation, whereas the interaction between the dose and time explained 70% (marginal R^2^). The effect of the time periods on the percentage change in the fGCM concentrations was dose-dependent (F = 9.055, df = 16, *p* < 0.001). The high-dose treatment revealed a peak at 24 h (160.54 ± 78.32) and 96 h (211.56 ± 33.07) post-injection (Figure 4B). Whereas, as with urine, the low dose and saline administration did not result in a peak value at any point in time (Figure 4B). As with the uGCM, there was variation in the raw baseline and response fGCM concentrations between individuals (Figure 5); however, individual variation explained 6% and 2% of the percentage change in the male and female fGCM concentrations, respectively.

## 4. Discussion

Five enzyme immunoassays were successfully validated for measuring the GCM concentrations in naked mole-rat urine and feces. The formerly established EIA by Edwards [16], 5α-pregnane-3β,11β,21-triol-20-one EIA detecting GCMs with a 5α-3β-11β-diol structure [36], was the most suitable assay for analyzing the urine samples of the males. For the female urine samples, the most suitable EIA was an 11-oxoaetiocholanolone EIA detecting GCMs with a 5β-3α-ol-11-one structure [10]. For quantifying the GCMs in the fecal samples, the best-performing EIA for both sexes was the 11-oxoaetiocholanolone EIA detecting 11,17-dioxoandrostanes [34], which was not tested by Edwards et al. [16]. The naked mole-rat females only showed a response post-ACTH challenge in feces (for high-dose individuals), whereas most males showed a peak post-injection in both tested matrices. Due to the limited sample size, the inconclusive response of the females when analyzing urine, and the logistical advantage of using the same assay between sexes when analyzing the fecal samples, we recommend using feces as a hormone matrix to determine the GCM concentrations in naked mole-rats.

To our knowledge, only five ACTH challenge test studies have so far investigated dose-dependent responses (Table 3). Yet, studies of this nature could be useful to investigate if the severity of a stressor is reflected in the matrix of interest by the best-performing EIA. However, thus far, there have been conflicting results. For cattle (*Bos taurus taurus*), Palme et al. [41] did not find a significant correlation between doses and peak concentrations in blood or feces; however, the peak fGCM concentration for the higher doses occurred later. For mice (*Mus musculus f. domesticus*), Touma et al. [3] found that for both males and females, the peak values during a low-dosage experiment were significantly lower than the peak values during the high dose, indicating that the respective EIA of choice was able to reflect dosage-dependent effects. For Eastern rock sengis (*Elephantulus myurus*), Medger et al. [42] found that females showed a significant increase in fGCM concentrations after receiving a high dose of synthetic ACTH, but there was no significant difference in uGCM concentrations when comparing the low- and high-dose treatments. On the other hand, the authors also found that male uGCM concentrations were significantly higher during the high-dose treatment compared to the low- and saline-dose treatment; however, there was no difference in fGCM concentrations in the high, low, and saline treatment [42]. Interestingly, for female black-tailed prairie dogs (*Cynomys ludovicianus*), Crill et al. [43] found that dose-dependent differences could be detected in plasma; however, in feces, no dose-dependent differences were seen, and the authors speculated that this could potentially be related to gut transit time.

Our study found sex- and dose-dependent variations in the response to ACTH. In the analyzed male naked mole-rat urine samples, a peak in the GCM concentrations can be seen at 6 h post-injection in the animals who received either a low or high dose of ACTH. Urine GCM concentrations returned to the baseline after 24 h. Whereas in the male fecal samples, there was a peak at 12 h post-injection for the low dose, and a peak at 24 h for the high dose. Overall, the results here show the fGCM concentrations went below a 100% increase at 72 h and returned to the baseline after 96 h post-injection. The naked mole-rat male response to the doses is similar to the findings in cattle [41], whereby there were no significant differences between the peaks from the two doses; however, the higher dose peak occurred later. Only the female naked mole-rats who received a high dose of ACTH showed a peak in the uGCM and fGCM concentrations after 12 and 24 h post-injection, respectively. The uGCM concentrations returned to the baseline after 24 h, whereas the fGCM concentrations did not return to the baseline during the experiment, possibly due to an unrecorded stressor. This is in line with what Medger et al. [42] found in female Eastern rock sengi fGCM samples whereby only females that received the high dose showed a peak in GCMs for the analyzed urine and fecal samples.

In all, it is not uncommon to see sex-specific differences in the non-invasive monitoring of stress-related hormones [23,25,44]. For example, in black-tufted marmosets (*Callithrix penicillate*), results showed that females had an overall higher response to ACTH and saline administration compared to males [45]. Furthermore, it is possible that the differences detected in response to a perceived stressor do not necessarily reflect quantitative differences in GCM output between the sexes, but rather sex-related differences in the composition of hormone metabolites reflected due to different cross-reactivity levels in the respective antibody in the EIA [22]. For example, in a study on laboratory rats (*Rattus norvegicus*), the 5α-pregnane-3β,11β,21-triol-20-one EIA was successfully validated for both sexes; however, high-performance liquid chromatography revealed that corticosterone metabolites differ between the sexes [46]. Finally, there are possible differences in the route of excretion between the sexes; for example, in bank voles (*Myodes glareolus*), evidence suggests that the main excretory route of corticosterone is feces for males, whereas in females, excretion of corticosterone is evenly distributed between urine and feces [47]. Overall, the sex-specific differences may be due to physiological differences in the metabolism and route of excretion, and/or differences due to life history traits of the species. Our results suggest that female naked mole-rats responded differently to the dose-dependent ACTH treatment than males. In addition, the male naked mole-rats showed a response to the saline treatment in both matrixes. Edwards [16] found no significant sex-related differences in the metabolism of cortisol and in the route of excretion. Based on one life history trait facet, it is possible that female naked mole-rats may be adapted to high-stress conditions due to the breeding female being the most aggressive member of the colony and thus asserts her dominance through aggressive behaviors such as shoving [48]. Conversely, males are possibly more sensitive to HPA activation and thus respond to stressors such as handling. However, this is speculative and further studies that replicate the radiometabolism study conducted by Edwards et al. [16], replicate saline, high, and low doses of ACTH treatments, and conduct focal observations of antagonistic behaviors by the queen and determine GCM correlates for the recipients would help to understand the differences in the sexes’ responses to a high and low dose of ACTH in the species.

The determined peak GCM values post-administration for both sexes in both matrixes (6 h and 24 h) appear similar to the findings of Edwards et al. [16]. The difference between the peak time in the two matrixes for both sexes is possibly due to the excretion time within the species [49]. However, given the average size (42 g) of the naked mole-rat individuals in this study, the time to reach peak fGCM concentrations is relatively long [16]. For example, in the similar-sized (~40 g average mass) Cairo spiny mouse (*Acomys cahirinus*), laboratory mice, and meadow vole (*Microtus pennsylvanicus*), the peak in the fGCM concentrations occurred ~5 h [50], ~8 h [3], and ~10 h [51] post-ACTH injection, respectively. Thus, our results support Edwards et al.’s [16] suggestion that naked mole-rats may have a relatively long gut transit time, possibly due to their water retention adaptations [28,52].

The role of stress In cooperatively breeding species such as naked mole-rats is under much debate [53,54,55]. Our results suggest preliminary evidence that the role of stress in naked mole-rat colonies is possibly sex-dependent. Whereby, since the male naked mole-rat responded to both low and high ACTH doses, it might suggest that in a colony setting, males respond the same to both high and low stressors. In the females, they may only potentially respond to highly stressful events, such as colony disruptions, but do not respond to relatively low stressful events such as antagonistic behaviors from the queen. Interestingly, Blecher et al. [56] found that circulating plasma cortisol concentrations were higher in separated individuals when compared to within colony measurements and when paired for both males and females. Since these individuals have been separated for an extended period, this may lead to endocrine exhaustion from chronic stress [12]. This is possibly due to the individual variation in the baseline values (Figure 3 and Figure 5). Further studies are required to investigate the difference in fGCM concentrations between breeding and non-breeding male and female naked mole-rats to understand the potential sex-related differences in responses to a high- and low-dose ACTH challenge.

Unexpectedly, for both sexes, the results show there were some individuals for which there was no peak in the fGCM concentrations detected by the chosen 11-oxoaetiocholanolone EIA, but there was a peak in the uGCM concentrations and vice versa. A possible explanation for this observation could be changes in steroid metabolite excretion. Changes in steroid metabolite excretion via urine have been linked to starvation [57] and sodium intake [58,59]. Starvation here is unlikely since the animals are given food *ad libitum*. Thus, these results could be a potential indication of high sodium concentrations [59] or dehydration. During naked mole-rat urine sample collection, it was relatively common to see small pellets of what we assume was allantoin (T. Majelantle, per obs.). However, these results must be interpreted with caution and further investigation on the route of excretion of metabolites in naked mole-rats, and how it relates to their nutrient and water intake from their diet, is required. Although inconclusive and preliminary, this finding highlights the importance of validating the effectiveness of EIAs in both urine and feces, instead of one matrix only.

An important limitation of this study is that individuals were not treated with both the high and low dose of ACTH, which makes dose-dependent responses more difficult to interpret due to potential individual variability in stress perception. Having all individuals treated with the high and low dose would have been useful in better understanding dose-dependent responses in the species. Thus, a study with more individuals (minimum six for each sex and a randomized administration of all treatments for each individual) will be useful to elucidate if the individual differences are due to dose, sex, or other factors which have not been considered.

## 5. Conclusions

This study has identified suitable EIAs for the non-invasive measurement of stress-related hormone concentrations in naked mole-rats. Based on the results presented here, we recommend using fecal samples when quantifying GCMs in the species. Speculatively, the results here suggest that there are possible sex-related differences in GCM output in relation to dose-dependent ACTH treatment. The limitations of this study are the relatively low numbers of animals investigated. More research using a larger sample size is required to understand the underlying mechanisms which cause the differences in male and female responses to high and low doses of ACTH.

## Figures and Tables

**Figure 1 animals-13-01424-f001:**
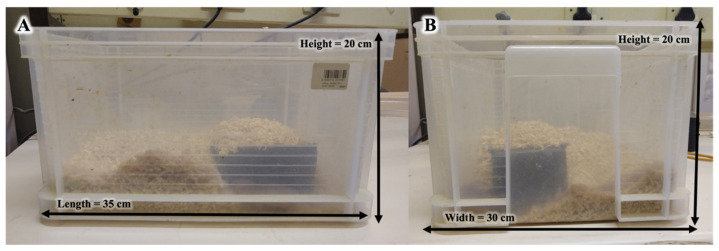
Photographs of side (**A**) and frontal (**B**) view of the naked mole-rat (*Heterocephalus glaber*) disperser plastic chambers.

**Figure 2 animals-13-01424-f002:**
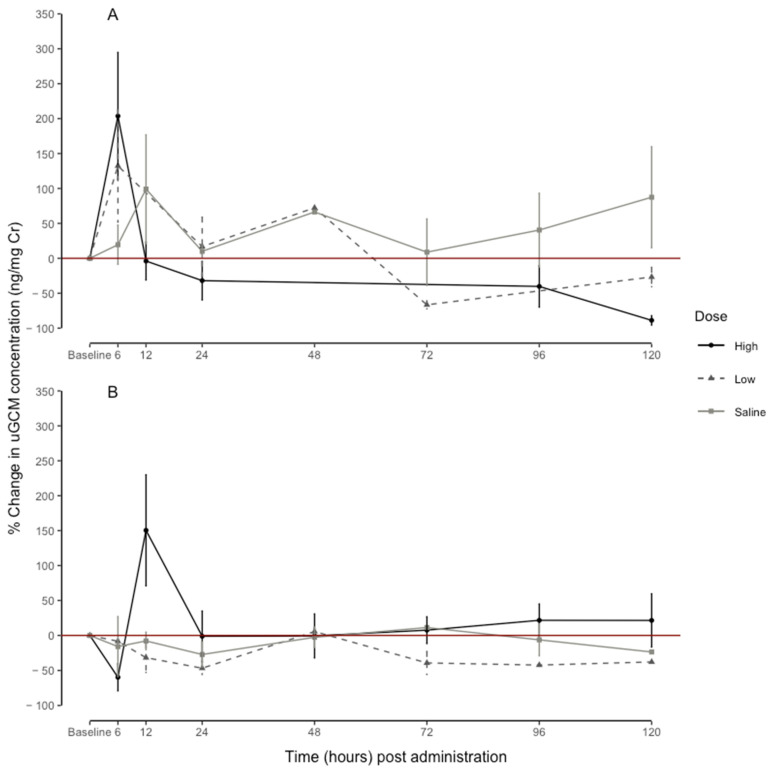
Overall peak increase (%) in urine glucocorticoid metabolite concentrations (uGCM, ng/mg creatinine) in relation to individual baseline (before and 48 h after administration) to 5 male (**A**) and 5 female (**B**) naked mole-rats (*Heterocephalus glaber*) during a high dose of ACTH (60 μg/100 g body mass), low dose of ACTH (20 μg/100 g body mass), and saline (200 µL sterile physiological saline: control) for each time interval. Points = mean, whiskers = standard error.

**Figure 3 animals-13-01424-f003:**
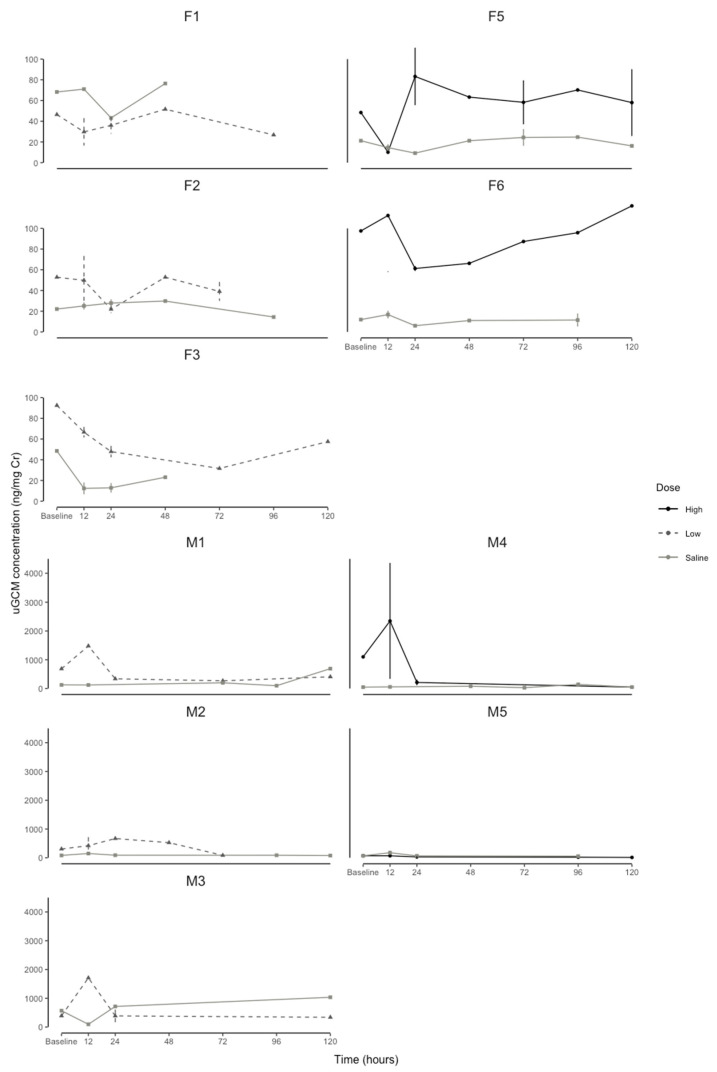
Individual naked mole-rat (*Heterocephalus glaber*) urine glucocorticoid metabolite concentration (uGCM, ng/mg creatinine) profiles of baseline (before and 48 h after administration) high dose of ACTH (60 μg/100 g body mass), low dose of ACTH (20 μg/100 g body mass), and saline (200 µL sterile physiological saline: control) for 5 males (M1, M2, M3, M4, M5) and 5 females (F1, F2, F3, F5, F6) at each time interval. One male (M6) and one female (F4) excluded due to small sample size. Points = mean, whiskers = standard error.

**Figure 4 animals-13-01424-f004:**
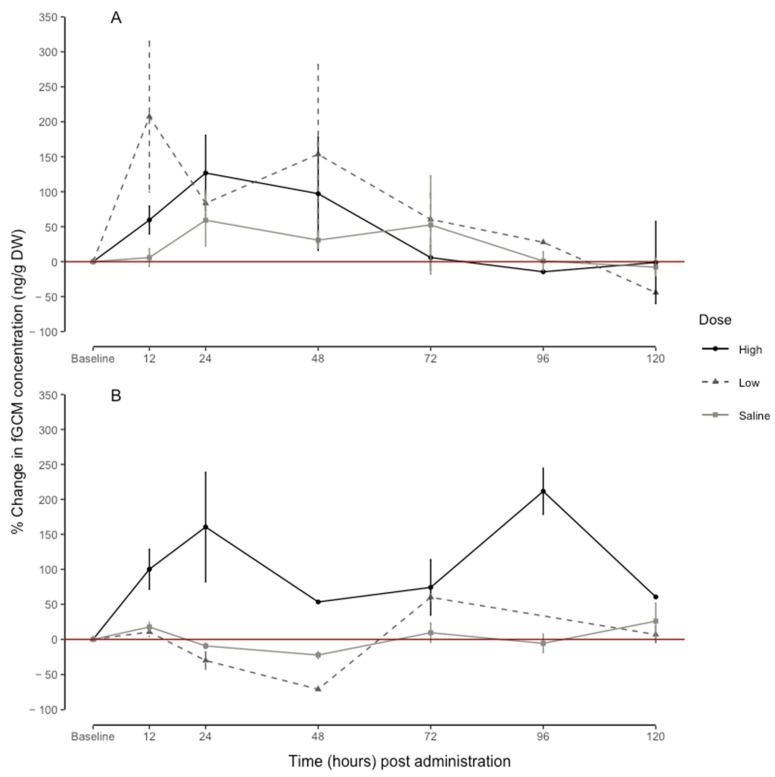
Overall peak increase (%) in fecal glucocorticoid metabolite concentrations (fGCM, ng/g DW) in relation to individual baseline (before administration) of 6 male (**A**) and 6 female (**B**) naked mole-rats (*Heterocephalus glaber*) during high dose of ACTH (60 μg/100 g body mass), low dose of ACTH (20 μg/100 g body mass), and saline (200 µL sterile physiological saline: control) for each time interval. Points = mean, whiskers = standard error.

**Figure 5 animals-13-01424-f005:**
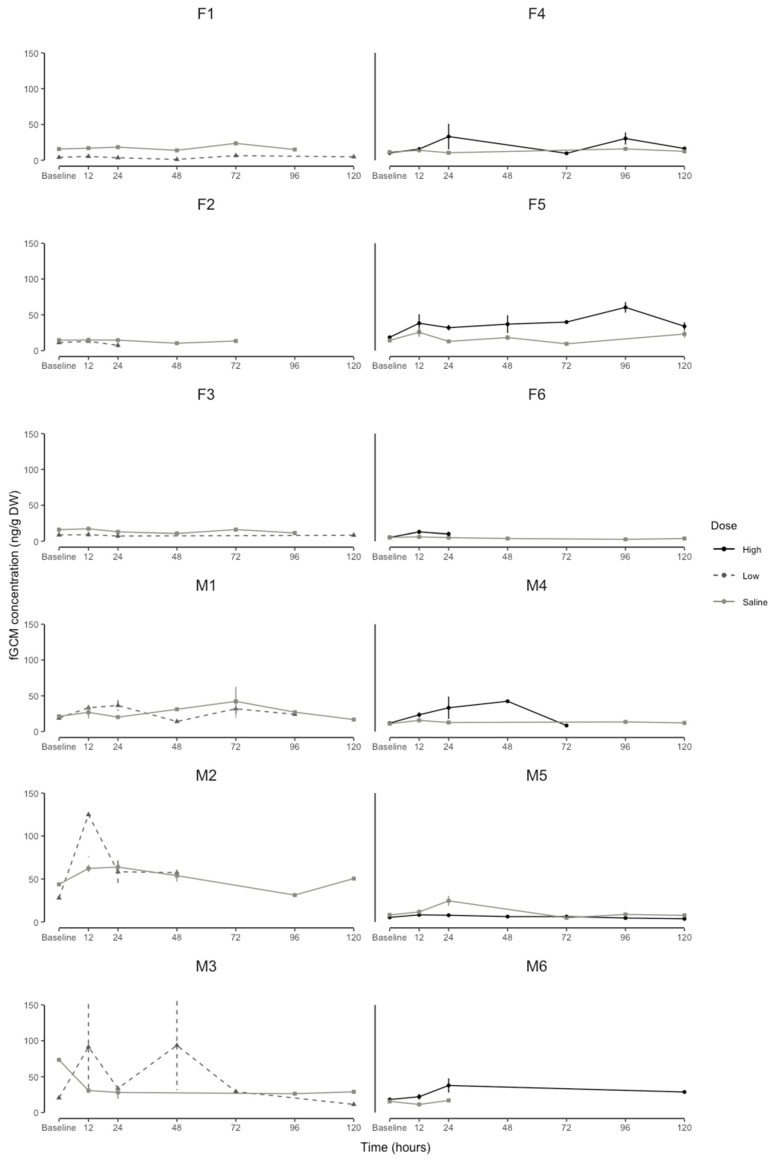
Individual naked mole-rat (*Heterocephalus glaber*) fecal glucocorticoid metabolite concentration (fGCM, ng/g DW) profiles of baseline (before administration) high dose of ACTH (60 μg/100 g body mass), low dose of ACTH (20 μg/100 g body mass), and saline (200 µL sterile physiological saline: control) for 6 males (M1, M2, M3, M4, M5, M6) and 6 females (F1, F2, F3, F4, F5, F6) at each time interval. Points = mean, whiskers = standard error.

**Table 1 animals-13-01424-t001:** Individual peak increase (%) in urine glucocorticoid metabolite concentrations (uGCM, ng/mg creatinine) within the first 48 h post-injection for 5 male and 5 female naked mole-rats (*Heterocephalus glaber*) which received a low (20 ug/100 g) or high (60 ug/100 g) dose in relation to individual baseline (EIA specific median of all values before and 48 h after administration).

Dose	ID	EIA				
		Cortisol	Corticosterone	5α-Pregnane	Oxoaetio	Oxoaetio
					Cholanolone I	Cholanolone II
Low	F1	−42	−7	40 ^	**174 ^**	11
	F2	60 ^	−23 *	46 *	26 *	38 *
	F3	**325**	**193**	19	**130 ***	−23 *
High	F5	29	−13	−15	−5	**170**
	F6	**106 ***	**105 ***	−2 *	28 *	71 *
Low	M1	17 ^	69 ^	**115 ^**	−5 ^	26
	M2	4 *	−58 *	**131 ***	88 *	88 *
	M3	**1545 ^**	**457 ^**	**338 ^**	**424 ^**	**367 ^**
High	M4	**261**	91	**295 ^**	73 ^	71
	M5	**995 ^**	**612 ^**	**113 ^**	91 ^	90 ^

Numbers in bold indicate percentage change >100, ^ indicates that the peak occurs within 6 h post-administration, and a * within 6–12 h post-administration.

**Table 2 animals-13-01424-t002:** Individual peak increase (%) in fecal glucocorticoid metabolite concentrations (fGCM, ng/mg DW) within the first 48 h post-injection for 6 male and 6 female naked mole-rats (*Heterocephalus glaber*) which received a low (20 ug/100 g) or high (60 ug/100 g) dose in relation to individual baseline (EIA specific median of all values before administration).

Dose	ID	EIA				
		Cortisol	Corticosterone	5α-Pregnane	Oxoaetio	Oxoaetio
					Cholanolone I	Cholanolone II
Low	F1	36 *	36 *	−10 *	76 *	−2 *
	F2	38	73 *	83	**195**	**218 ***
	F3	29 *	−1 *	−12 *	64 *	−8 *
High	F4	79	16	13	**392**	−5
	F5	74	27 *	19	**160**	38
	F6	73	**110**	**120 ***	**233 ***	−29
Low	M1	**230**	77	7	**315**	−1 *
	M2	**166**	64	87	**172 ***	**97**
	M3	54	8	−5	65	27
High	M4	56 *	80	40	**309**	9 *
	M5	99	94	*64	**147**	68 *
	M6	**296**	92	39	**274**	72

Numbers in bold indicate percentage change >100 and a * indicates within 6–12 h post-administration.

**Table 3 animals-13-01424-t003:** Validation studies which investigated dose-dependent response in ACTH challenge test and respective species.

Species	Dose	EIA	Citation
Cattle (*Bos taurus taurus*)	0.25 mg, 0.5 mg, and 1 mg	11-oxoetiocholanolone I	[41]
Mice (*Mus musculus f. domesticus*)	20 μg and 60 μg/100 g	5α-pregnane-3β,11β,21-triol-20-one EIA	[3]
Black-tailed prairie dogs (*Cynomys ludovicianus*)	* 4 μg and 12 μg/100 g	Cortisol- and Corticosterone- EIAs	[43]
Eastern rock sengis (*Elephantulus myurus*)	` 20 μg and 60 μg/100 g	5α-pregnane-3β,11β,21-triol-20-one EIA	[42]

* Reported as 4 IU and 12 IU/kg, ` Reported as 0.2 and 0.6 μg/g.

## Data Availability

Data supporting the reported results will be sent by the corresponding author upon request.

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
