# Peer review of "Validation of Enzyme Immunoassays via an Adrenocorticotrophic Stimulation Test for the Non-Invasive Quantification of Stress-Related Hormone Metabolites in Naked Mole-Rats"

_animals, 2023, doi:10.3390/ani13081424_

Round 1

Reviewer 1 Report

Summary: This manuscript describes the validation and comparison of five enzyme immunoassays for quantifying glucocorticoid metabolites in the urine and feces of male and female naked mole rats, using high or low-dose ACTH compared to a saline control injection. This is a useful study, and an excellent approach to set a solid foundation for future work exploring glucocorticoids non-invasively as they relate to animal welfare. There are six males and six females in the study, three of each receiving the saline control followed by the low dose, and three of each the saline control followed by the high dose ACTH challenge. There is variation in responses between individuals, sexes, matrices and potential dose-depended responses, but overall the authors suggest optimal EIAs for male urine, female urine and feces from both sexes, and that fecal sampling may provide the more reliable method for future work.

General comments:

Individual variation does complicate interpretation with no clear temporal response that is consistent across all individuals, doses or sample types; because individuals received either high or low dose, not both, it is difficult to distinguish between individual differences in response due to dose or other factors. The use of statistical modelling does help to account for this to determine general responses, but the variation is an interesting factor that isn’t discussed very much. One area that I feel should be discussed is the GCM response to saline injection – in the males (based on average profiles in Figures 1 and 2), uGCMs and fGCMs look to be increased compared to baseline throughout the data collection period – could this be due to the handling or sampling procedure? This could also influence the other treatment groups so should be mentioned in the discussion, and could also be related back to the suggested sex differences in responsiveness to external stressors. Overall I think this is a nice manuscript, but I think a bit more discussion of the variation in responses that can’t be fully explained by sex or dose effects could be beneficial to stimulate future work to increase understanding of this area.

Specific comments:

1. Please check the consistency of capitalization for the corticosterone, cortisol and 11- oxoaetiocholanolone EIAs throughout – sometimes these are capitalized and sometimes not. Creatinine is also often capitalized in table/figure legends when it needn’t be.

2. Please check the consistency faeces/faecal to feces/fecal to maintain consistency throughout

Line 48: please add an apostrophe to ‘naked mole rat’s unique characteristics’
Line 62: please add a comma before ‘which’

Line 90: please hyphenate dose-dependent

Line 111: do you mean 5 days after the start of sample collection?

Line 116: please provide the explanation for uGCM and fGCM on the first use of this abbreviation (which is given later on lines 152-153)

Lines 131-132: can you comment on how you avoided urine contamination of feces?

Lines 186-188: I think I misunderstood this when I first read it - do you mean the period starting at 48-hours post ACTH challenge was used to calculate a baseline (as opposed to within 48-hours)? It might help to clarify by stating the hours included, i.e. ‘the median of all samples collected prior to the challenge (hours -120 to 0) and post-challenge (hours 48-120) were used to calculate baselines….’

Lines 189-192: it might be useful to summarize the number of samples obtained for other animals to help the reader see why these two in particular have been excluded – 8 samples for the ACTH period doesn’t sound that low (although the saline control would need to be excluded for this individual), and based on Figure S1, there are other individuals included with relatively low sample sizes, so it would be useful to clarify what the threshold for exclusion was.

Lines 192-3: What do you mean here by no decrease in fGCM concentrations post-injection? Do you mean no decrease in the 48-120 hours time-frame (if my interpretation above is correct)? It would be helpful to clarify this statement.

Line 220: Do you mean ACTH-injection specifically (as is mentioned for female below), or any injection?

Table 1 and results interpretation: In table 1 it shows that the cortisol and corticosterone EIAs showed peaks in urine for 2/5 females (comparable with the 11-oxoaetiocholanolone I EIA), but these are not mentioned in the results section – it would be useful to include these, and why these assays were not considered for further analyses.

Table 1 and Table 2: It would be useful to indicate which animals received the low and the high dose here to help with interpretation – this is listed in supplementary table 1, but would be useful to include in the main manuscript

Table 1 and 2 legends: please italicize Heterocephalus glaber

Line 282: should this refer to Figure 2A as it is still the male data?

Line 349: please change ‘which’ to ‘that’

Line 352, 358: please italicize species

Line 355: please remove the ‘s’ from ‘a perceived stressors’

Line 358-9: your point could be strengthened here by adding whether the same assay was used for both sexes, did they validate separately by sex?

Line 361: please change ‘suggest’ to ‘suggests’

Line378/9: I would suggest changing ‘gut passage time’ to ‘excretion time’ here, because you are discussing both fecal and urinary routes

Line 379/80: This sentence would read better if changed to ‘However, given the average size (42 g) of naked mole-rat individuals in this study, the time to reach peak fGCM is relatively long’

Line 383: please add a comma before ‘respectively’

Figures S1 and S2 – I personally would like to see these profiles in the main manuscript instead of in supplementary material, where many readers might not look for them. I think recognizing that individual variation in responses can be quite significant is important to a better understanding of applicability of GCMs. I would also suggest paneling the figures so that all low dose are on the left and high dose on the right (female at the top, male at the bottom) would help to visualize patterns in the data. The figure for M2 fGCM is also on a different axis to the other males, it would be good to have those consistent.

Reviewer 2 Report

Dear Authors,

Thank you for the opportunity to review the manuscript ‘Non-invasive quantification of stress-related hormone metabolites in the naked mole-rat (Heterocephalus glaber)’.

The study assessed the suitability of five EIAs to monitor the physiological response of twelve captive-bred naked mole-rat to a low dose and a high dose of ACTH as well as a dose of saline solution. The research focused a detecting cortisol metabolite in both urine and feces.

Comments:

Title

‘Non-invasive quantification of stress-related hormone metabolites in the naked mole-rat (Heterocephalus glaber)’ the title needs to mention somehow that the study is about an ACTH challenge to evaluate most suitable EIAs to quantify stress related hormone metabolites.

Introduction:

Line 58 ‘Glucocorticoids (GCs) are a part of multiple physiological processes’. Please add maintaining homeostasis, at least, as one of the multiple processes. If you can, it would be great to have more details for the untrained reader.

Line 74 ‘In addition, using feces and urine as a hormone matrix over plasma…’ plasma was not one of the matrices mentioned in line 72. Please add.

Line 81 together with ref 22 please add Santamaria et al. 2021 ‘Identification of Koala (Phascolarctos cinereus) Faecal Cortisol Metabolites Using Liquid Chromatography-Mass Spectrometry and Enzyme Immunoassays’

Line 83 Together with ref 23 please add Santamaria 2023 The Effect of Disease and Injury on Faecal Cortisol Metabolites, as an Indicator of Stress in Wild Hospitalised Koalas, Endangered Australian Marsupials

Materials and Method

General comment

The information on when the samples were taken, how many samples and what type of analysis was done needs to be displayed in a clearer manner (i.e a table) currently it is very confusing to understand and follow the full procedure. In particular, under ‘2.5. Enzyme Immunoassays’ there is ample description of sample size and collection times which should be described under Sample collection.

 Line 101 Please add a picture of the chambers to make it clear to the reader. The description doesn’t specify if these chambers were made of transparent plastic. Could the animals sense each other through smell or sounds? Was the study carried out during reproductive season? Please mention this and what steps were taken to avoid any effect of these interactions on u or fGCM values. This is very important especially for the results of the saline injection.

 Line 105 Grammar: Move ad libitum before ‘every second day’

Line 112 onwards ‘Subsequently, all animals received either a low (20 ug/100g body weight) or a high (60 ug/100g body weight) dose of a synthetic ACTH solution 25 days later.’ How many got the low and how many the high? Or did all the animals get both high and low? There is no justification to why the high and low doses ere used and the introduction does not investigate which other studies have used high and low doses and for which purpose (I have seen mentioned in the discussion, but it needs to be introduced somewhere).

 Line 116 ‘To establish individual baseline uGCM and fGCM concentrations, urine and fecal samples were collected five days prior to- and five days post- saline and ACTH administration, respectively.’ Is it at day 5 before and after or for five days before and after? It is not clear if it is a one off collection or a continuous collection of samples for the total 10 days.

 Line 120 ‘were frequently’ Can you please give an idea of average frequency during the time of the study or were all defecations collected during the study?

 Line 122 ‘the animals’ please replace ‘the’ with ‘each’.

 Line 129 Please explain how feces were separated from urine to avoid incorrect fGCM values due to contamination. As freeze-drying only removes water not metabolites, if the feces are in urine the concentration of GCMs may be misleading.

 2.5. Enzyme Immunoassays As explained earlier, this section needs a re-write to make it clearer and some information needs to be moved to other sections.

 Line 148 I am not sure if this is samples meaning defecation or part of a defecation and why such difference in sample numbers between individuals?

 Lines 164-165 (12– 22 November 2022, 10 days total) should it be 2020?

 Results

 Table 1 and Table 2 please add a row above the first row and insert ‘EIAs’, remove the acronym EIA from the row with the EIA names, this avoids repetitions.

 Discussion

 Line 297 Start with ‘Five’

 Line 314 ‘However, thus far, there have been conflicting results. For cattle (Bos taurus taurus), Palme et al. [39] did not find significant correlation between doses and peak concentrations in blood or feces’ could this be due to the species-specific negative feedback loop response? Maybe something to mention? Also referring to the method Line 112, if only some animals were injected with high and some with low doses, could there be individual differences as well? After all physiological responses to stress hormones are individual.

 Line 352  Callithrix penicillate and any other scientific names in italics

 Line 359 Please add Santamaria et al 2021 (Seasonal Variations of Faecal Cortisol Metabolites in Koalas inSouth East Queensland) and 2023 (The effect of disease and injuries on faecal cortisol metabolites, as an indicator of stress in wild hospitalised koalas, endangered Australian marsupials to ref 44.

 Please add a section with Limitations before the conclusion

Round 2

Reviewer 2 Report

Dear Authors,

Thank you for making the requested changes to your manuscript.

Looking at the table with the list of EIAs, the 11-oxo aetiocholanolone is only shown as oxoetiocholanolone and the word is split as oxoaetio and cholanolone. I think that if possible, the name should be 11-oxo and below etiocholanolone so to split the name where it should be split according to its nomenclature (please see https://www.scbt.com/p/11-oxo-etiocholanolone-739-27-5 as an example).

Aside from this beautification, congratulations for a job well done.